# Population Structure, Genetic Diversity and Candidate Genes for the Adaptation to Environmental Stress in *Picea koraiensis*

**DOI:** 10.3390/plants12061266

**Published:** 2023-03-10

**Authors:** Ya Wang, Zeping Jiang, Aili Qin, Fude Wang, Ermei Chang, Yifu Liu, Wen Nie, Cancan Tan, Yanchao Yuan, Yao Dong, Ruizhi Huang, Zirui Jia, Junhui Wang

**Affiliations:** 1State Key Laboratory of Tree Genetics and Breeding, Research Institute of Forestry, Chinese Academy of Forestry, Beijing 100091, China; 2Key Laboratory of Forest Ecology and Environment of National Forestry and Grassland Administration, Ecology and Nature Conservation Institute, Chinese Academy of Forestry, Beijing 100091, China; 3Forestry Research Institute in Heilongjiang Province, Harbin 150081, China

**Keywords:** *Picea koraiensis*, genetic differentiation, geographic isolation, climatic factors, heavy metal stress, molecular mechanisms

## Abstract

*Picea koraiensis* is major silvicultural and timber species in northeast China, and its distribution area is an important transition zone for genus spruce migration. The degree of intraspecific differentiation of *P. koraiensis* is high, but population structure and differentiation mechanisms are not clear. In this study, 523,761 single nucleotide polymorphisms (SNPs) were identified in 113 individuals from 9 populations of *P. koraiensis* by genotyping-by-sequencing (GBS). Population genomic analysis showed that *P. koraiensis* was divided into three geoclimatic regions: Great Khingan Mountains climatic region, Lesser Khingan Mountains climatic region, and Changbai Mountain climatic region. Mengkeshan (MKS) population on the northern edge of the distribution area and Wuyiling (WYL) population located in the mining area are two highly differentiated groups. Selective sweep analysis showed that MKS and WYL populations had 645 and 1126 selected genes, respectively. Genes selected in the MKS population were associated with flowering and photomorphogenesis, cellular response to water deficit, and glycerophospholipid metabolism; genes selected in the WYL population were associated with metal ion transport, biosynthesis of macromolecules, and DNA repair. Climatic factors and heavy metal stress drives divergence in MKS and WYL populations, respectively. Our findings provide insights into adaptive divergence mechanisms in *Picea* and will contribute to molecular breeding studies.

## 1. Introduction

*Picea* is the third-largest genus in Pinaceae and a major component of coniferous forests in the northern hemisphere. Most conifers have the characteristics of long generations, large population size [1,2,3], and a low level of interspecific reproductive isolation [4]. Spruce underwent rapid differentiation during the Pliocene, Qinghai–Tibet Plateau uplift, and Quaternary glacial oscillation [5]. The continuous gene flow among sympatric and parapatric species makes the evolutionary relationship more complex [6,7]. Geographic isolation and gene introgression are two possible driving factors of species differentiation [8,9]. In conifers with long generations, a long period of time can elapse from the onset of differentiation to the cessation of gene flow. Studies have suggested that gene flow between different species continues to decline over time when the species is under either isolation or selective pressure [10]. Understanding gene introgression and selection pressure is of great significance for further understanding spruce phylogeny.

*Picea koraiensis*, belonging to the genus *Picea*, is distributed in the northeast of the Great Khingan Mountains, Lesser Khingan Mountains, Zhang Guangcai Mountains, Wanda Mountains, and Changbai Mountains in China, with elevations ranging from 300 m to 1600 m. As an independent species, *P. koraiensis* has generally been accepted by the academic community, but its population genetic structure and geographical migration and spread history are not clear. There are different hypotheses about the origin of the genus spruce, the “North American Origin Theory” [11,12,13] and the “Asian Origin Theory” [14,15,16], which are supported by different DNA evidence. Whether spruce originated in North America or southeastern Asia, its migration route is through the northeast plain of China, from North America to the Qinghai–Tibet Plateau, or from east Asia or the Qinghai–Tibet Plateau to North America. Therefore, the distribution area of *P. koraiensis*, the Chinese northeast plain, is an important transition zone for spruce migration. Many fossils of the genus Picea from the Jurassic to Early Cretaceous, which is the earliest fossil of the genus Picea found thus far, have been found in the distribution area of *P. koraiensis* [17,18,19]. Some studies have been performed on the natural population division and genetic diversity of *P. koraiensis*. For example, Zhang et al. [20] divided the natural population of *P. koraiensis* into five climatic zones. The genetic diversity of the *P. koraiensis* population was analyzed by phenotypic biosystematics, seedling morphology, growth, biomass, allozyme variation [20], mineral element content [21], and random amplified polymorphic DNA (RAPD) molecular markers [22]. Variations within and among populations of *P. koraiensis* have been found. The variation within populations was greater than the variation among populations, and the genetic distance of *P. koraiensis* was positively correlated with geographical location to a certain extent. The level of genetic differentiation among different geographical populations of *P. koraiensis* is very high, but the population genetic structure, driving factors, and molecular mechanism of differentiation are not clear, and the phylogeography of *P. koraiensis* at the whole genome level has not been reported.

In this study, *P. koraiensis* was used as the research object to study the genetic structure of the natural population and the adaptive differentiation mechanism of different geographical populations to environmental stress by genotyping-by-sequencing. Based on the above research, we put forward the following questions: (i) What is the genetic relationship, genetic diversity, and differentiation of *P. koraiensis* in the natural distribution area? (ii) Is there any obvious differentiation among populations in different geographical distribution areas of *P. koraiensis*? (iii) If there is obvious differentiation, what is the molecular mechanism of differentiation motivation and adaptability?

## 2. Results

### 2.1. Characterization of GBS-Seq Data and SNPs

The genotyping-by-sequencing (GBS) of 127 samples produced the original reads of 563.64 Gb. After quality filtering, the clean reads of 532.685 Gb were retained and the average value of each individual was 4.19 Gb. The average efficiency was 94.5%. The sequencing results showed that the average Q20 value, Q30 value, and GC content were 97.38%, 91.63%, and 40.21%, respectively. After comparing the filtered data with *P. abies*, the average comparison rate was 95.33%. The number of SNP (single nucleotide polymorphism) bits detected was 10,375,590. Phylogenetic analysis was conducted with 127 samples, and a total of 523,761 SNPs with missing ratio ≤20% minor allele frequency (maf) ≥0.01 were identified.

### 2.2. Phylogenetic and Population Genetic Structure Analysis

According to the phylogenetic tree constructed by the neighbor-joining (NJ) method (Figure 1), with *P. jezoensis*, *P. likiangensis*, and *P. pungens* as outgroups, 112 individuals of *P. koraiensis* were divided into 3 groups: Great Khingan Mountains group (Group I), Changbai Mountain group (Group II), and Lesser Khingan Mountains group (Group III). Among these groups, Group I is composed of the Mengkeshan (MKS) population in the northern Great Khingan Mountains, which is located at the base. Group II is composed of four populations: Hailin (HL), Muling (ML), Linjiang (LJ), and Tianqiaoling (TQL) populations, and Group III is composed of the Wuyiling (WYL), Aihui (AH), Gaofeng (GF) and Zhanhe (ZH) populations. Group II and Group III are sister clades. The results of the phylogenetic tree showed that the genetic relationship of different populations of *P. koraiensis* was related to geographical distribution. The population genetic structure was further analyzed by ADMIXTURE v1.23 software [23]. The number of clusters (K) was set from 2 to 10, and the best number of clusters was selected according to the minimum cross-validation rate (Figure 2). The optimal number of clusters was three. When K = 2 to 4, the error rate of cross-validation was small, indicating that 2 to 4 clusters is more reasonable. When K = 2, 127 samples were divided into *P. koraiensis* and the outgroup, and the MKS population of *P. koraiensis* had a small amount of outgroup gene infiltration. When K = 3, *P. koraiensis* was divided into 2 groups, the WYL population and other populations. When K = 4, other populations had extensive gene flow except for the WYL and GF populations. The results showed that gene introgression occurred in the MKS population and outgroups, and the WYL population had greater differentiation from other populations. To further analyze the population genetic structure, 112 individuals of *P. koraiensis* were analyzed by principal component analysis (PCA) (Figure 3). The results showed that *P. koraiensis* could be divided into three main clusters: MKS, WYL, and other populations. The results of PCA and admixture analyses indicated that there is an obvious differentiation between the two populations (MKS and WYL) and others.

### 2.3. Population Diversity and Selective Sweep Analysis

The data are shown in Table 1; the π(nucleotide diversity) of the *P. koraiensis* population was 0.00194 to 0.00231. Among the populations, the π value of the WYL population was the lowest (0.001943), while the π value of LJ was the highest (0.00235). The *F_st_* (fixation index) among *P. koraiensis* was 0.0254 to 0.1569, among which the mean *F_st_* between the WYL population and other populations was the highest (0.1211), followed by the MKS population (0.0983), and the mean *F_st_* between the TQL population and other populations was the lowest (0.0538). This result is consistent with the phylogenetic analysis, indicating that the WYL and MKS populations have a high degree of differentiation from other populations, and these two populations may be isolated from other populations and have less gene flow, resulting in a significant decrease in nucleotide diversity. That is, these two populations experienced a high degree of selection in the process of evolution. The degree of differentiation of the LJ, TQL, and ZH populations was low, and the three populations had frequent gene flow with other populations and high nucleotide diversity. The Tajima’s D values of the nine populations of *P. koraiensis* were all positive (mean 0.1554), indicating that these nine populations may have experienced balanced selection.

The main purpose of selective sweep analysis was to find the selected genes of the population, which further reveals the adaptive mechanism of population evolution. In our study, four populations of Changbai Mountain group were used as background group. The MKS and WYL populations were used as test groups, respectively. Fixation index (*F_st_*) and nucleotide diversity ratios (π ratio) were used in combination to identify regions where selective sweeps occurred during species differentiation. A total of 645 and 1126 selected genes were identified in the MKS and WYL populations, respectively (Appendix A).

The results showed that the selected regions of the MKS population were *F_st_* (MKS) ≥ 0.19 and π ratio ≥ 1.57 (Figure 4a). Based on GO (gene ontology) enrichment analysis, the molecular functions of the genes were concentrated mainly in the response to radiation, response to light stimulus, macromolecule methylation, response to reactive oxygen species, nitrogen compound metabolic process, ubiquitin-dependent protein catabolic process, and nucleic acid metabolic process (Figure 5a; Appendix A). As shown in Figure 6a, the KEGG (Kyoto Encyclopedia of Genes and Genomes) pathways were concentrated mainly in ABC transporters, RNA transport, inositol phosphate metabolism, glycerophospholipid metabolism, RNA polymerase, nucleotide excision repair, mismatch repair, flavonoid biosynthesis, cysteine and methionine metabolism, peroxisome, endocytosis, and biosynthesis of amino acids. Among these genes, we found genes related to photoperiod, flowering, and photomorphogenesis (e.g., CUL4, GATA9, SBH2, and MBD9), and genes involved in the cell response to water scarcity (such as RPK2, CYT1, and MGL), which may be related to the climate adaptation of the MKS population to the northern Great Khingan Mountains. In the selected region, we also found genes related to metal ion transport (e.g., ABCB family genes), flavonoid biosynthesis-related genes (e.g., ANS, DFR), cysteine metabolism-related genes (e.g., DNMT1, DNMT3A), amino acid biosynthesis genes (e.g., GlyA, thrB, cysK, RPE, and TYRAAT), peroxidase metabolism-related genes (e.g., AGXT, SOD1, and PEX1), genes related to response to reactive oxygen species (e.g., HSP, SODCC, CPN60-2, and CYT1), signal transduction-related genes (e.g., CTR1), and oxidative stress defense-related genes (e.g., MYB family genes), showing that the MKS population may have been subjected to stress. In addition, genes related to DNA repair (e.g., CUL4, MNAT, RFA1, MLH3, and POLD2) were also found, suggesting that the MKS population may have DNA damage under adverse conditions.

In the WYL population, the selected regions were *F_st_* (WYL) ≥ 0.32 and π ratio ≥ 2.5 (Figure 4b). Based on the GO enrichment analysis, the molecular functions of the selected genes were concentrated mainly in chromosome condensation, regulation of macromolecule metabolic process, gene expression, cation transport, RNA metabolic process, metal ion transport, ATP hydrolysis coupled proton transport, etc. (Figure 5b, Appendix A). As shown in Figure 6b, the KEGG pathways were concentrated mainly in phosphonate and phosphinate metabolism, phagosome, glycerophospholipid metabolism, oxidative phosphorylation, ribosome, RNA polymerase, peroxisome, glutathione metabolism, proteasome, etc. Among these genes, we found genes related to plant–pathogen interactions (e.g., CNGC, CPK, FLS2, NOA1, CERK1, CML, RPS2, MPK6, and KCS), response to salt-stress-related genes (e.g., CYT1, FAR4, MYB78, and PPR40), bivalent metal ion transport-related genes (e.g., TL17, VHA_D, SAUR32, AGT1, TIP2-1, and GLR 3.3), univalent metal ion transport-related genes (e.g., HAK family genes, YMF19, VHA, AVP1, and OEP64), plant hormone signal-transduction-related genes (e.g., IAA, SAUR, ARR-B, GID1, MPK6, and EBF1_2), flavonoid-biosynthesis-related genes (e.g., ANS, FLS, ANR, and LAR), and DNA-repair-related genes (e.g., TOP3, RFA1, BRCA2, MLH1, PMS2, and RBX1), showing that the WYL population may differentiate under the stress of environmental factors, and the metal ions in the soil of its distribution area may be higher than the metal ions in the soil of other population areas.

## 3. Discussion

### 3.1. Geographical Isolation, Climate Heterogeneity, and Gene Introgression

In our study, whole-genome SNPs were obtained by the GBS-seq technique, and the phylogeny, population structure, and genetic differentiation of *P. koraiensis* distributed in northeast China were analyzed. The results showed that the genetic relationship among different populations of *P. koraiensis* was consistent with the division of geographical and climatic regions: there was gene introgression between the MKS population and outgroup, and there was obvious genetic differentiation among the MKS and WYL populations and other populations. Climate heterogeneity, gene introgression, and geographical isolation are the main reasons for the differentiation of *P. koraiensis*.

Climatic factors such as temperature and precipitation are likely to be among the important factors driving the differentiation of *P. koraiensis* due to different habitats. The phylogenetic results divided *P. koraiensis* into three groups corresponding to three geoclimatic regions: the Great Khingan Mountains climatic region, the Lesser Khingan Mountains climatic region, and the Changbai Mountain climatic region. The climate of the northern Great Khingan Mountains area is characterized by long sunshine, low precipitation, and low temperature; the climate of the Changbai Mountain region is characterized by medium sunshine, medium precipitation, and medium temperature; and the climate of the Lesser Khingan Mountains area is characterized by short sunshine, low precipitation, and low temperature [20]. Zhang et al. [20] divided the distribution of *P. koraiensis* in China into five sections: the Changbai Mountains area, the Laoye Mountain–Wanda Mountains area, the Lesser Khingan Mountains area, the north part of Great Khingan Mountains, and the southwest part of Great Khingan Mountains. Among these sections, the spruce species distributed in the southwestern part of the Great Khingan Mountains is *P. meyeri var. mongolica*. The Laoye Mountain–Wanda Mountains area belongs to the branches of Changbai Mountain, with little difference in climate factors. The results of the phylogenetic and genetic diversity analyses showed that the populations of the Laoye Mountain–Wanda Mountains and Changbai Mountain areas were clustered into monophyletic groups. Therefore, we classified the two populations in the Laoye Mountain–Wanda Mountains (HL and ML populations) in the Changbai Mountain climatic zone. Among other species, there are also examples of temperature and moisture influencing population differentiation, such as Liaodong oak [24], eucalyptus in Australia [25], and alpine conifers in the western Alps [26].

Gene flow and introgression are also important causes of intraspecific differentiation in *P. koraiensis*. On the one hand, the size of gene flow between different groups within the species and the degree of differentiation were inversely proportional. These two groups, WYL and MKS, have little gene exchange with other groups, low nucleotide diversity, and greater differentiation; the ZH, TQL, and LJ populations have extensive gene flow with other groups, high nucleotide diversity, and low differentiation. On the other hand, gene introgression between species promotes the differentiation of different populations within *P. koraiensis*. The results of phylogenetic and population differentiation analysis showed that the MKS population was more differentiated from other populations and had a small genetic component of outgroups. Because of the large overlap in distribution between the MKS population of *P. koraiensis* and its sympatric species *P. jezoensis*, we speculate that there is a small amount of gene introgression from *P. jezoensis* to the MKS population. Many closely related species of *P. koraiensis* have been successfully crossed with closely related species of *P. jezoensis* in hybridization experiments [15]. Fossil evidence suggests that during the Pleistocene, *P. jezoensis* and its extinct relatives were widespread in northeastern Russia [27], that the ancestors of *P. koraiensis* and its relatives may have appeared in east Asia during the same period [28], and that the ancestors of *P. jezoensis* and *P. koraiensis* underwent infiltration before *P. koraiensis* and its relatives’ migration to Eurasia [29]. Gene introgression between other species of the genus spruce has been reported repeatedly: *P. glauca*, *P. engelmannii*, *and P. sitchensis* in western North America [30,31,32,33,34,35] and *P. likiangensis*, *P. purpurea*, and *P. wilsonii* in the Qinghai–Tibet Plateau [6,7,36,37,38]

Geographical isolation is an important reason for the intraspecific differentiation of *P. koraiensis*. At the northern edge of the *P. koraiensis* distribution area, the MKS population has the characteristics of geographical and genetic isolation because of spatial isolation and limited gene communication; that is, the differentiation characteristics of the MKS population accord with the central–marginal hypothesis, in which marginal populations show lower genetic variation and higher differentiation than central populations. There are similar examples in other species, such as *Abies sachalinensis* in Hokkaido and Pinus cembra L. in the Alps [39,40]. The WYL population is distributed in the Lesser Khingan Mountains gold mining area. Due to the influence of mining and different geological conditions, the natural environment shows fragmentation and islanding, forming an ecologically isolated island, which is different from other distribution areas of *P. koraiensis*. Furthermore, there is geographical isolation. There are similar examples of other species distributed in mining areas, such as Caryophyllaceae [41,42], *Scopelophila cataractae* [43], and *Colocasia esculenta* [44]. In summary, geographical isolation leads to the differentiation of species and the formation of obvious geographical differences to a great extent; the same is true of Lamiaceae and Circaeasteraceae [45,46].

### 3.2. Molecular Mechanisms of Population Differentiation

To further detect natural selection signals in different populations of *P. koraiensis* and explore the differentiation mechanism, we performed selection sweep analysis to identify candidate genes based on KEGG and GO enrichment analyses involving various biological functions, using populations of the Changbai Mountain group as background populations and the more differentiated MKS and WYL populations as control populations. The key functional genes for which differentiation between different populations of *P. koraiensis* was subject to selection were to improve the resistance of the population, to promote flowering and fruiting ability for adaptation to different habitats, and to continuously reproduce offspring.

The MKS population is distributed in the northern part of the Great Khingan Mountains and is geographically distant from other populations, with large climatic differences. Climatic heterogeneity drives significant differentiation of the MKS Mountain population, and the function of genes shows the molecular adaptation mechanism of the MKS population to its habitat. The northern Great Khingan Mountains have long sunshine hours (approximately 2600 h per year and 2300 h per year in the Changbai Mountain region), and light affects the photoperiod, circadian rhythm, and photomorphic building of plants. The results of our selection sweep analysis showed that 15 genes in the MKS population were enriched in response to radiation, and 14 genes were enriched in response to light stimulus. Among these genes, CUL4 in Arabidopsis can regulate the flowering time in a short period of time during the day and is involved in photomorphogenesis [47,48]; GATA9 is involved in the regulation of circadian rhythms in Arabidopsis [49]; SBH2 is involved in photomorphogenesis [50]; and MBD9 regulates photoperiod and flowering in Arabidopsis through DNA methylation [51,52]. The northern Great Khingan Mountains are characterized by low annual precipitation (463 mm average annual rainfall and 680 mm average annual rainfall in Changbai Mountain), and our results show that 5 of the genes are enriched in response to water deprivation, with RPK2 being involved in pollen germination, response to cold, and response to water deprivation [53]; CYT1 being involved in defense responses to bacteria, hormone homeostasis, and salt stress [54,55]; and MGL being involved in cellular response to sulfate starvation and cellular response to water deprivation [56]. The northern Great Khingan Mountains are characterized by low temperatures (−2.4 °C annual mean temperature and 3.2 °C annual mean temperature in Changbai Mountain), and organisms living in this environment face various growth-related challenges from low temperatures, such as reduced lipid membrane fluidity. The lipid bilayer transmits external signals to its interior and protects the integrity of various organelle membranes in unfavorable environments. The results showed that 86 genes were associated with membrane-bound organelles and that 6 genes were involved in glycerophospholipid metabolism. Among these genes, PLC is involved in leaf membrane lipid remodeling [57] and regulation of stomatal movement and root development under abiotic stress [58,59]. Remodeling of membrane lipids may protect membranes from low temperatures. We hypothesize that these genes may play a major role in maintaining membrane integrity under low-temperature conditions. The functions of these genes suggest that MKS populations have evolved a range of genes adapted to the long light hours, low annual rainfall, and low temperatures of the habitat.

Among the Lesser Khingan Mountains group, the WYL population is distributed in the gold mining area, and there are also large amounts of iron ore and agate in its distribution area. The WYL population is distributed at a low altitude (average elevation approximately 350 m), and due to tailings pollution caused by mining, monsoon effects, and rainfall leaching, many heavy metals, such as Cd, Cr, Cu, Ni, Pb, and Zn, in the soil exceed background values, especially Pb, Cd, Zn, and Cu [60,61]. Heavy metals cause oxidative damage, disruption of cellular homeostasis, DNA strand breaks, fragmentation of proteins or cell membranes, and damage to light and pigments [62]. Plants undergo a series of physiological changes in response to heavy metal stress to adapt to their environment. The first line of defense of plants against heavy metal stress is to block the transport of metal ions in the roots through the plasma membrane [63]. The enrichment analysis showed that 19 genes are involved in the transport of metal ions, of which 5 genes were enriched in ABC transporter, where the ABC transporter family (e.g., ABCB1, ABCA15, ABC1, ABCG28, and ABC1K8) is involved in the efflux of metal ions from the plasma membrane and in the resistance of plants to heavy metals and aluminum toxicity [64,65]. For example, the ABCB family genes enhance resistance to cadmium and lead in Arabidopsis [65], and ABCB1 is upregulated in mango when it is subjected to lead stress [66]. After heavy metals penetrate the first line of defense and enter the cell, plants tolerate or neutralize the toxicity produced by heavy metals through the synthesis of biomolecules such as metal chelators and specific amino acids [63]. Forty-eight genes were identified as regulators of cellular macromolecular biosynthetic processes, among which PCYT2, PCYT1, EPT1, and PHO1 are involved in the synthesis of phosphate and hypophosphate [67,68,69,70,71,72,73]; ANS, FLS, and ANR are involved in flavonoid biosynthesis [74,75,76]; CysCK is involved in the synthesis of cysteine [77]; and G6PD, SPE3, GST, and RRM1 are involved in glutathione synthesis [78,79,80,81]. Phosphates, flavonoids, cysteines, glutathione, and other phytochelators are markers of heavy metal stress in plants [82,83]. GST genes are upregulated in horse rushes, tomatoes, and radishes when they are subjected to heavy metal stresses such as cadmium [84,85,86]. When both of the above mechanisms against heavy metal stress are exhausted, plants activate oxidative stress defense mechanisms and the synthesis of stress-related proteins and signaling molecules [63]. Seven genes were associated with oxidative stress defense in plants, among which MPK6 and FLS2 were enriched in the MAPK pathway and IAA, SAUR, and GID1 were enriched in the hormone signaling pathway. In Arabidopsis, MPK6 can be induced by Cd ions and Cu ions [87,88,89,90], and the IAA (AUX) concentration increases to promote lateral root growth and protect plants from Cd [91]. In addition, the regulation of transcription factor family expression by heavy metal stress has been reported [92,93,94]. The transcription factors identified in this study are members of the MYB family (MYB4, MYB52, MYB58, MYB78), WRKY family (WRKY7, WRKY35, and WRKY42), NAC, and BZIP68. In Arabidopsis, MYB family genes such as MYB4 are highly induced by Cd and Zn metal stress [95]. The expression of WRKY family members was significantly elevated under Cu and Cd metal exposure [96]. BZIP transcription factor expression was also induced by Cd stress [97]. For DNA damage caused by heavy metals, plants initiate DNA damage repair mechanisms with six genes involved in DNA repair, of which RBX1 and RFA1 are involved in nucleotide excision repair [98,99], MLH1 and PMS2 are involved in mismatch repair [69,100,101], and TOP3 and BRCA2 are involved in homologous recombination [102,103], suggesting that the WYL population evolved multiple DNA repair pathways. In addition, compared with Changbai Mountain, the sunshine in the Lesser Khingan Mountains area is short and the temperature is low. The WYL population also identified genes related to light morphogenesis, circadian rhythm, and cold tolerance. For example, in *Arabidopsis thaliana*, the CSN4 and CSN8 participate in the development of seedlings in the dark and promote the development of light morphology in the dark [104,105]; ABC1K8 and BLH1 are involved in regulating circadian rhythms [106,107]; and LSM1A, CIPK9, and PSRP2 are involved in plant adaptation to cold environment [108,109,110]. Our results show that the genome of the WYL population not only responds to the climatic environment of the Lesser Khingan Mountains area, but also may be stressed by heavy metals in mining areas, and the genome has undergone adaptive evolution.

## 4. Materials and Methods

### 4.1. Sample Collection and GBS Analysis

We collected samples according to the 4 climatic zones of *P. koraiensis* and identified a total of 9 populations of 10 to 15 individuals each, for a total of 112 individuals (Figure 7, Table 2). A total of 2 populations of the Changbai Mountain climate zone (Region I): Tianqiaoling (TQL) and Linjiang (LJ); 2 populations of the Laoye Mountain, Zhangguangcai Mountain, and Wanda Mountains climate zone (Region II): Muling (ML) and Hailin (HL); 4 populations of the Lesser Khingan Mountains climate zone (Region III): Wuyiling (WYL), Zhenhe (ZH), Aihui (AH), and Gaofeng (GF); and 1 population of the Great Khingan Mountains climate zone (Region IV): Mengkeshan (MKS). Three species, *P. jezoensis* (sympatric species of *P. koraiensis*), *P. likiangensis*, and *P. pungens*, were used as outgroups, with five individuals per species. Plants from each population were separated by at least 100 m at the time of sampling, and needles were dried directly using silica gel and kept at low temperature. DNA extraction from plant needles was performed using a modified bromocetyltrimethylamine (CTAB) method, followed by a Nanodrop-1000 spectrophotometer (Nanodrop, MA, USA) to detect nucleic acid concentration and DNA purity on 1% agarose gel electrophoresis.

DNA samples were qualified, sequenced, and constructed in GBS libraries according to the methods reported by Poland et al. [111]. Each sample extracted from 1.5 μg of DNA was digested in 96-well plates with EcoRI and NiaIII restriction enzymes. The digestion reaction was performed in 96-well plates using 100 ng of DNA for each individual reaction digested with EcoRI and NIaIII (New England Biolabs, Ipswich, MA, USA). The digested product was mixed with A1 and A2 linkers at 25 pmoL per well, adding linkers at both ends of the DNA. Library pooling, selection of size (400–600 bp) on a 1% agarose gel, column purification using PCR purification kit (NEB), and amplification of 12 cycles using Phusion DNA polymerase (NEB) were carried out. After the second column wash, the average fragment size was estimated on a BioAnalyzer 2100 (Agilent, Santa Clara, CA, USA) using a DNA1000 chip, and library quantification was performed using PicoGreen (Invitrogen, Carlsbad, CA, USA). Sequencing with PE125 on HiSeq4000 (Illumina, San Diego, CA, USA) adjusted the common library to 10 nmol. Numerous raw sequencing sequences (raw reads) were obtained through high-throughput sequencing. To ensure data quality, the quality control of the original data should be carried out before information analysis, and data filtering should be used to reduce data interference information. We filtered the dismounted raw reads more strictly to obtain high-quality clean reads for subsequent information analysis. The conditions for filtering were as follows: (1) reads containing adapter sequences were removed; (2) reads containing unknown base N ratios greater than 10 were removed; and (3) low-quality reads (the number of bases of mass value Q ≤ 10 accounted for more than 50% of the entire read) were removed for genomic electronic digestion assessment. To check the efficiency of digestion and analyze all markers, we digested the reference genome according to the site of restriction enzymes and counted the corresponding conditions for later evaluation. Sequencing evaluation: statistics are made of output data, including sequencing data yield, sequencing error rate, Q20 content, Q30 content, N content, GC content, etc., to evaluate whether the library construction and sequencing are successful.

Using the comparison software Burrows-Wheeler Aligner (BWA) ver.0.7.12 (Li, Cambridge, UK) [112], the filtered reads were compared with the reference genome (European spruce) by the mem algorithm, and the alignment parameter was-k 32-M. After the comparison, the results were marked by Picard (1.129) software. The Unified Genotyper module of the software GATK (3.446) is used to detect the variant of multiple samples of the processed comparison files, and the detected variations are filtered by VariantFiltration. The filtering parameters are -Window 4, -filter “QD < 4.0 || FS > 60.0 || MQ < 40.0”, -G_filter “GQ < 20”. ANNOVAR was used to make functional comments on the detected variant. Using VCFTOOL v.0.1.11 (Danecek, Cambridge, UK) [113], the SNP loci (127 individuals) obtained by GBS sequencing were filtered if the missing ratio ≤20% dint maf ≥0.01 and all indels were removed.

### 4.2. Phylogenetic and Population Structure Analyses

The phylogenetic tree of 127 individuals was constructed by the Neighbor-Joining method and the P-diatance model [114] based on MEGA version X software [115] and corrected by Jukes-Cantor, Tajima-Nei, and the Maximum Composite Likelihood model. A bootstrap consensus tree was formed from a sample of 1000 replicates. iTOL version 6.4 was used to edit and beautify the phylogenetic tree [116].

To carry out the cluster analysis of the population structure, the parameters of the ancestry coefficient matrix Q and the population allele frequency matrix F and K were calculated by admixture software [23] and finally plotted by the R software package (http://www.r-project.org/, accessed on 12 January 2023). The K value range was set from 1 to 100, and the default settings were used for the other parameters. The optimal clustering number was determined according to the minimum value of the cross-validation (CV) error, and the R language package was used to draw the results.

The genetic structure of the spruce population was evaluated by principal component analysis (PCA). We used GCTA software (http://cnsgenomics.com/software/gcta/pca.html, accessed on 12 January 2023) to reduce the dimension of the selected SNP loci and calculate the eigenvectors and eigenvalues. Three principal components were obtained by using GCTA, and the PCA distribution map was drawn by using the R language package. The values in the axis label parentheses represent the percentage of the principal component that explains the overall variance.

### 4.3. Genetic Diversity and Selective Sweep Analysis

Nucleotide diversity (π) (also known as θ π, Pi), genetic diversity analysis (Tajima’s D), and genetic differentiation among populations (*F_st_*) were analyzed by the PopGenome module in the R software package. According to the phylogenetic results, *P. koraiensis* was divided into three groups: the Changbai Mountain group, the Great Khingan Mountains group, and the Lesser Khingan Mountains group. Because of the moderate climate and frequent gene communication with other populations in the Changbai Mountain group, we took four populations (HL, ML, LJ, and TQL) in this group as background groups. Combined with the results of *F_st_*, the MKS population of the Great Khingan Mountains group and WYL populations of the Lesser Khingan Mountains group were used as control groups to carry out selective sweep analysis. We scanned the genome in 100 kb sliding windows with a step size of 10 kb to identify significant regions based on a fixed index (*F_st_*) and nucleotide diversity ratio (π ratio). We respectively screened the significant regions according to the top 5%, and the intersection region is the selected region. Genes in the selected region were defined as the selected gene, namely, the candidate gene. After obtaining the candidate genes in the selected region, the genes were analyzed by GO (gene ontology) (www.geneontology.org, accessed on 13 January 2023) and KEGG enrichment analysis. Then, the variations in SNPs in the coding region of the candidate genes were annotated, and the core functional mutations were speculated to provide clues for further verification.

## 5. Conclusions

In our study, phylogenetic, population structure and selective sweep analyses of different populations of *P. koraiensis* in northeastern China were performed by GBS-seq sequencing technology. The results of the phylogenetic analysis showed that *P. koraiensis* was divided into three geoclimatic zones: the Great Khingan Mountains climatic region, the Lesser Khingan Mountains climatic region, and the Changbai Mountain climatic region, with introgression in the MKS population and outgroups. Geographic isolation, climatic factors, and introgression are the driving factors for the differentiation of *P. koraiensis*. The MKS population at the northern edge of the distribution area and the WYL population in the mining area are the two populations with greater differentiation. The differentiation of the MKS population is consistent with the central–marginal hypothesis, and the results of selective sweep analysis show that the genes subject to selection removal in the MKS population are related mainly to flowering and photomorphogenesis, cellular response to water deficit, response to reactive oxygen species, flavonoid synthesis, and DNA repair. Climatic heterogeneity factors (e.g., sunlight, precipitation, and temperature) drive the differentiation of MKS; genes selected in the WYL population are related mainly to metal ion transport and macromolecule biosynthesis, such as flavonoids, phytohormone signaling, and DNA repair, and the WYL population may be subject to heavy metal stress to produce differentiation. Our results provide insights into adaptive evolution molecular mechanisms of genus spruce and contribute to spruce resistance improvement via genomic-assisted breeding.

## Figures and Tables

**Figure 1 plants-12-01266-f001:**
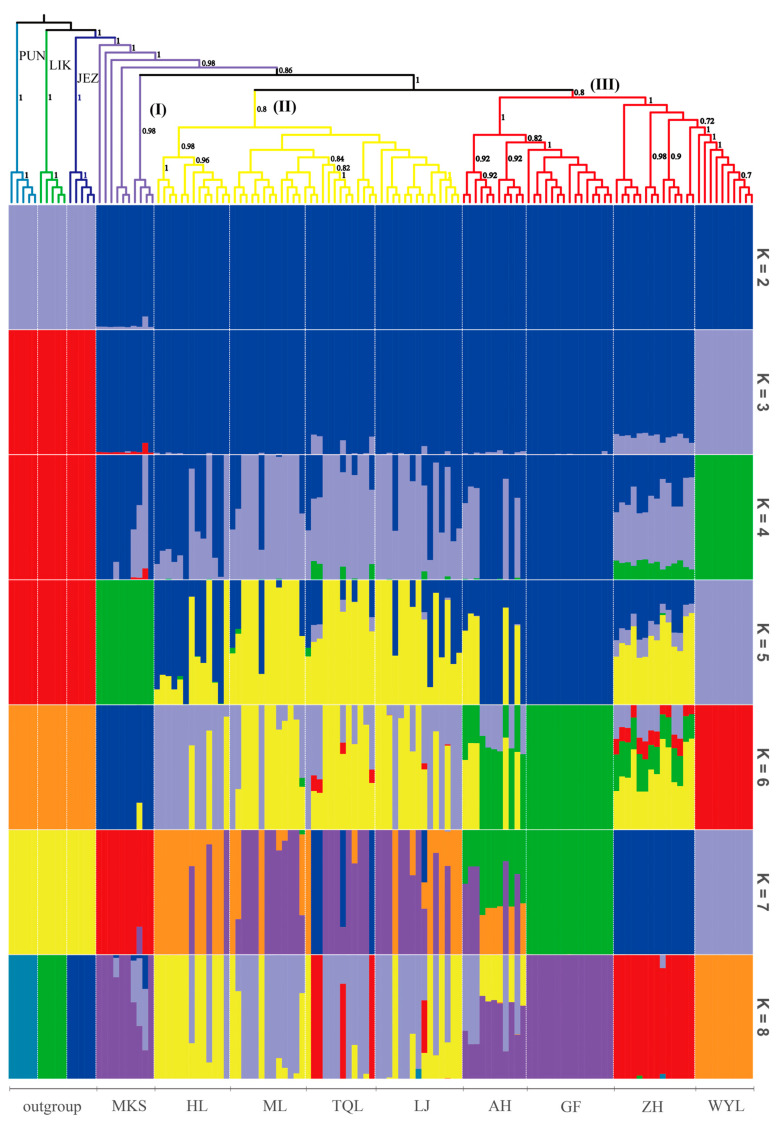
Phylogenetic tree and structure of 127 samples of *P. koraiensis* and outgroups with bootstrap value. PUN: *Picea pungens*; LIK: *Picea likiangensis*; JEZ: *Picea jezoensis.* I: Great Khingan Mountains group; II: Changbai Mountain group; III: Lesser Khingan Mountains group.

**Figure 2 plants-12-01266-f002:**
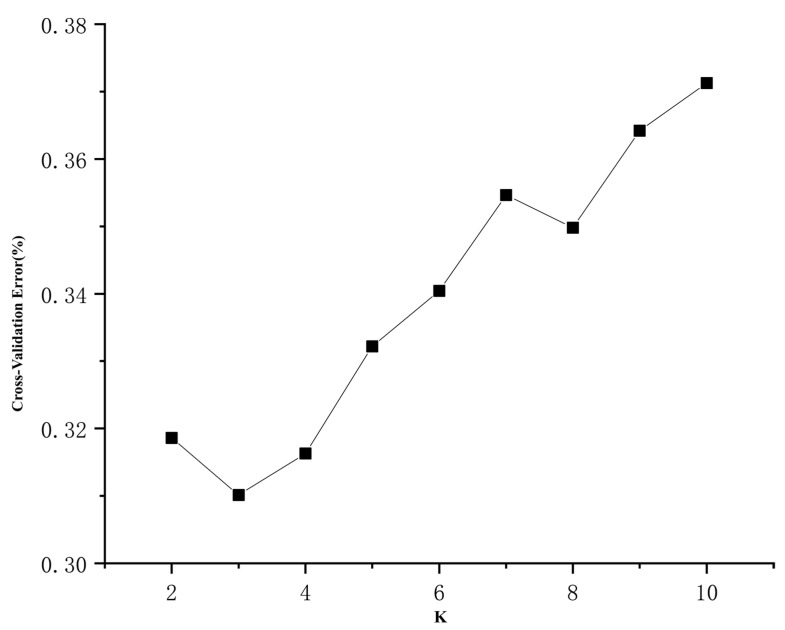
Cross-validation error value based on ADMIXTURE v1.23 software. The best number of clusters was 3 according to the minimum cross-validation rate.

**Figure 3 plants-12-01266-f003:**
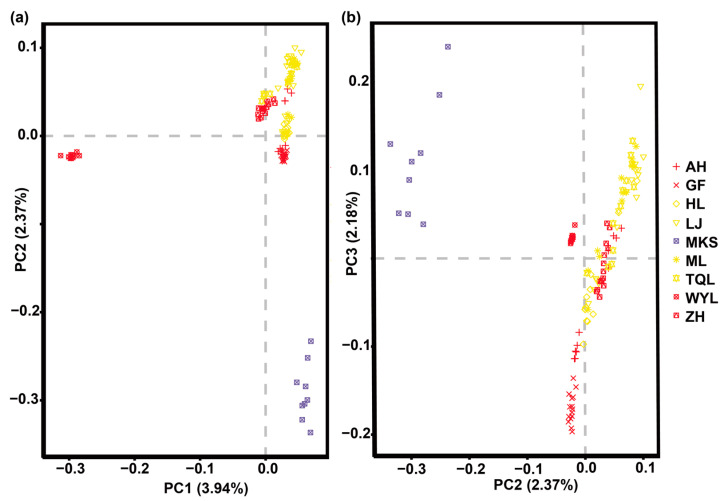
Principal component analysis of 9 populations of *P. koraiensis*. (**a**) PC1 and pPC2; (**b**) PC2 and PC3. PC: principal component.

**Figure 4 plants-12-01266-f004:**
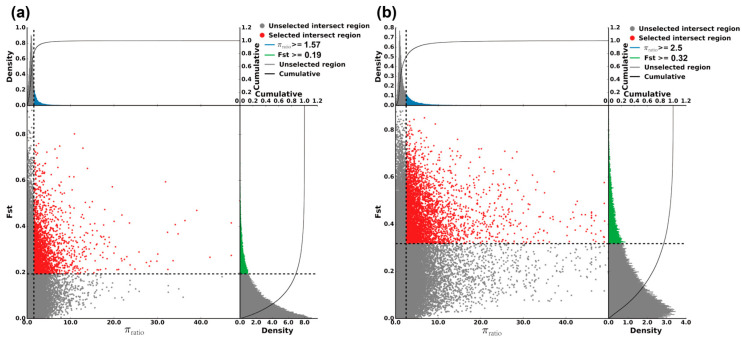
Genomic regions with selective sweep in MKS and WYL population: (**a**) MKS population; (**b**) WYL population.

**Figure 5 plants-12-01266-f005:**
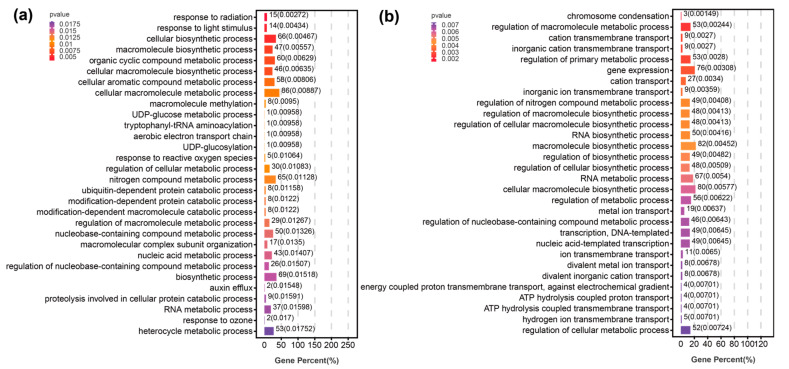
Annotation based on the GO database. The *Y*-axis indicates the top 30 GO terms, and the *X*-axis indicates the numbers of genes annotated to the terms and their P values: (**a**) MKS population; (**b**) WYL population.

**Figure 6 plants-12-01266-f006:**
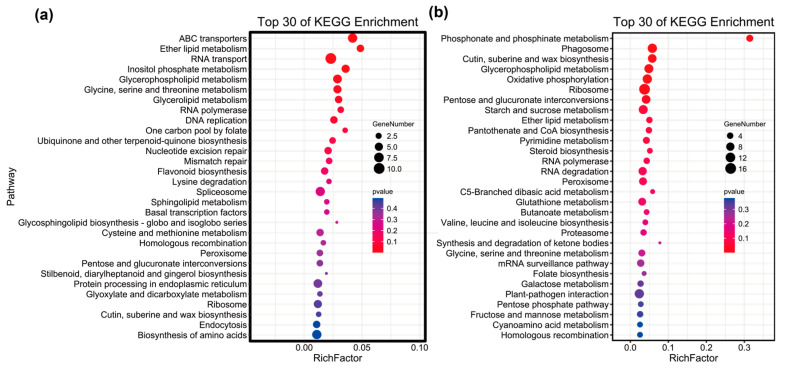
KEGG enrichment analysis of selected genes. The color scale indicates the *p* value: (**a**) MKS population; (**b**) WYL population.

**Figure 7 plants-12-01266-f007:**
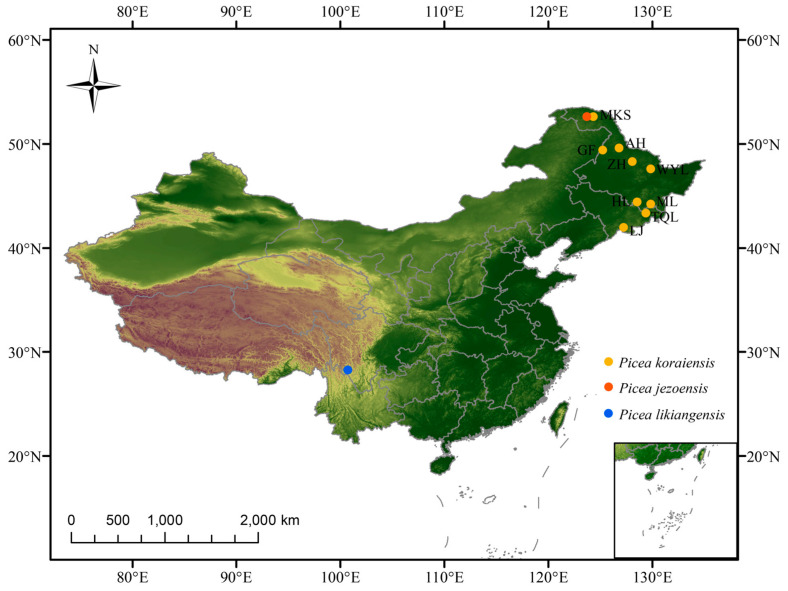
Geographical distributions of 9 populations of *P. koraiensis* with altitude as the base map.

**Table 1 plants-12-01266-t001:** *F_st_* between 9 populations, π and Tajima. D neutral test in *P. koraiensis*.

Population	GF	MKS	ML	HL	TQL	AH	WYL	LJ	ZH	π × 10^−2^	Tajima’s D
GF	0									0.2181	0.3892
MKS	0.1018	0								0.2285	0.1205
ML	0.0643	0.0878	0							0.2289	0.0526
HL	0.0628	0.0909	0.0399	0						0.2272	0.1307
TQL	0.0599	0.0838	0.0294	0.0371	0					0.2319	0.0297
AH	0.0529	0.0938	0.0500	0.0544	0.0462	0				0.2212	0.1657
WYL	0.1282	0.1569	0.1166	0.1182	0.1115	0.1191	0			0.1943	0.5110
LJ	0.0623	0.0873	0.0322	0.0399	0.0254	0.0488	0.1137	0		0.2351	0.0507
ZH	0.0491	0.0842	0.0408	0.0416	0.0371	0.0450	0.1042	0.0404	0	0.2314	0.1875

**Table 2 plants-12-01266-t002:** Locations of 9 *P. koraiensis* populations.

Population	NO.	Collection Site	Latitude	Longitude
GF	10	Mengke Mountain, Heilongjiang	52.63	124.31
MKS	13	Hailin, Heilongjiang	44.42	128.54
ML	15	Muling, Heilongjiang	44.21	130.21
HL	15	Linjiang, Jilin	41.99	127.24
TQL	13	Tianqiaoling, Jilin	43.36	129.37
AH	14	Wuyinling, Heilongjiang	48.67	129.42
WYL	14	Zhanhe, Heilongjiang	48.33	128.07
LJ	11	Aihui, Heilongjiang	49.62	126.81
ZH	15	Gaofeng, Heilongjiang	49.43	125.23

## Data Availability

The sequence reads associated with the paper are stored at the NCBI Sequence Read Archive in the BioProject PRJNA664562.

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
