# Peer review of "Population Structure, Genetic Diversity and Candidate Genes for the Adaptation to Environmental Stress in Picea koraiensis"

_plants, 2023, doi:10.3390/plants12061266_

Round 1

Reviewer 1 Report

Comments and Suggestions for Authors

The study carried out is aimed to study the genetic structure of the natural populations and the adaptive differentiation mechanism to environmental stress in genus Picea on the base of investigation of different geographical populations of Picea koraiensis by genotyping by sequencing. Picea is the third-largest genus in Pinaceae and a major component of coniferous forests in the northern hemisphere. The degree of intraspecific differentiation in it is high, but population structure and differentiation mechanisms are not clear. Therefore understanding gene introgression and selection pressure is of great significance for further understanding spruce phylogeny. P. koraiensis was used as the research object in the present study in order to clarify what is the phylogenetic relationship and population genetic structure in the natural distribution area of this taxon.

The manuscript follows the structure recommended from “Plants”. The research methods applied are appropriate and sufficient to achieve the objectives of the study. The results are well presented and supported by tables and figures that are of good quality.

The following recommendations can be made:

There are no given the meaning of all abbreviation used in the text, e.g. RAPD, SNPs, Fst, GO, KEGG, etc.

Materials and methods

The sentence in line 371-372 needs editing: “DNA extraction was extracted from plant needles using a modified bromocetyltrimethylamine (CTAB) method”, as follows: “DNA extraction from plant needles was performed using a modified bromocetyltrimethylamine (CTAB) method”

In line 398 “2)” have to be “3)”

Results

Line 103: “the error rate of cross-validation is” – “is” have to be changed to “was”

Line 111: PCA /the meaning have to be given here because is the first appearance of the abbreviation /

 Discussion:

Line 204:  climatic regions, there was gene introgression between the MKS population and …” – the “:” are more appropriate instead of “,”

 Line 468-469: “Climatic heterogeneity factors (e.g., sunlight, precipitation, and temperature) drive the 468 differentiation of MKS;” - the “:” are more appropriate instead of “;”

In conclusion, this manuscript is recommended for publication in “Plants”.

Reviewer 2 Report

Dear Authors,

The paper of  Junhui Wang and coauthors tilted Population Structure, Genetic Diversity, and Selective Sweeps 3 in Picea koraiensis revealed by single nucleotide polymorphism (SNP) markers describe genetic diversity patterns within P. koraiensis using SNP markers. This species was scarcely and exanimated using molecular markers and its evolutionary history is not fully resolved. The manuscript is well written but I highlighted a few points that need to be corrected and a few suggestions to better describe some issues.

Title: is too long and complicated maybe: genetic diversity and candidate genes for the adaptation,  etc.

Introduction:

- please, describe the geographic range of this species and include it in Fig. 7. This figure, especially populations also need the codes (like in Table S5) and designations to geographic regions because, for people from other parts of the world, it’s difficult to recognize them. Table S5 can be in the text not as an appendix.

- the first question of the author refers to the phylogenetic relationship of PK, but it is somewhat genetic relationship;  this question could be modified and connected with the second question like genetic diversity and differentiation …

 Results

- Fig. 1 – it is not clear what outgroup was included in the NJ tree, please complete it, using the species name; numbers I, II, and III were not described in fig. On the NJ tree, the bootstrap values were not shown. The cross-validation results of the clustering showed that when K = 3, the error rate was minimal, indicating that the optimal clustering number was exactly 3. This fact I would stress in the description of this fig.  Generally, all descriptions of figures should be completed. In Fig. 2,  please write what program and algorithm were used to construct cross-validation, and what K is optimal. Fig. 3, the font needs to be larger, I propose to use the figures but specify one color for each geoclimatic group, in this form the picture is not clear. Fig. 4, the font needs to be larger, Fig. 5 the font needs to be larger, and if the p values are larger than 0.1 (in the picture a) and 0.002-.007 in picture b, it means that in the second picture were shown only significant values. Therefore, I suggest ranking from highest to lowest values for genes, and also in picture a, where the values are not significant. Fig. 6, the font needs to be larger.

Lines 125-135. I do not understand the description of Table 1 in the text. Probably, the Fst values have moved between populations in this table. The π values are also different than in the text, it can be standardized.

Reviewer 3 Report

Authors need to revise because some problems are recognized in the article.

The first major question is the use of Picea abies as a reference genome.

Authors used the genome of another species, P. abies, as a reference for SNPs detection, but wondered whether this would result in a bias in the SNPs detected due to the influence of interspecies differentiation in the mapped regions.

 Also, we are doing GO enrichment analysis of the genes that are thought to be in the selective sweep candidate region, but I did not understand how they are identified as genes in the neighborhood region. It was difficult to identify the candidate genes based on the GBS sequence data alone, so I determined that the genes that were thought to be in the selective sweep candidate region were estimated based on the genome of P. abies. However, it was not clear to what extent the candidate regions were defined as neighborhoods. In addition, it is questionable whether the distribution of genes in P. abies and P. koraiensis should be considered the same.

I will point out some comments below in the article.

L87 "After the heterozygous individuals were removed, phylogenetic analysis was conducted with 127 samples, and a total of 523761 SNPs with missing ratio ≤ 20% maf ≥ 0.01 were identified."

Why exclude heterozygous individuals; wouldn't excluding heterozygous be assessed as over-differentiating the population?

L91-94 "According to the phylogenetic tree constructed by the neighbor-joining method (Figure 1), with P. jezoensis, P. likiangensis and P. pungens as outgroups, 112 individuals of P. koraiensis were divided into three clades: Great Khingan Mountains lade (Clade I), Changbai Mountain clade (Clade II) and Lesser Khingan Mountains clade (Clade III)."

Great Khingan Mountains lade (Clade I) is a paraphyletic group, and I think it is problematic to call it “clade”.

 L106-108 "When K=3, P. koraiensis was divided into two groups, the WYL population and other populations, and other populations had extensive gene flow except the GF population."

 This statement appears to be somewhat inadequate since gene flow is mentioned from the K=4 results, not just the K=3 results. Also, if you are going to mention gene flow, why not calculate the migulation rate, etc., instead of just the ADMIXTURE results?

L112-114 "MKS, WYL and other populations, consistent with the results of admixture analysis, indicating that there is an obvious differentiation between the 2 populations (MKS and WYL) and others."

I think the ADMIXTURE results alone are weak to state that the MKS is clearly different from the rest of the population, although an introgression from the outgroup can be seen.

 P5 Table1

Fst seems off, π is wrong (probably 10^-2 is missing)

L140-141 ” In our study, four populations of Changbai Mountain clade were used as background group.”

If the CladeII population is compared to the WHY population, it would include not only the specificity of the WHY population but also the differentiation between CladeII and CladeIII.

L146~

Although the gene in the selective sweep region is estimated, were no SNPs actually found in the gene portion? And if there were, wouldn't it be better to compare synonyms and nonsynonyms to detect selections?

P11 Figure 7

Please add the name of the populations to the map to make it easier to understand their location.

L418-420 “The SNP genotype was used to calculate the genetic distances between the 127 individuals using the p-distance method [107], and phylogenetic and molecular evolutionary analyses based on the neighbor-joining method were conducted using MEGA version X [108].”

You use p-distance to calculate distances, but don't you use JC69 model, K80 model or other corrections?

Round 2

Reviewer 2 Report

Dear Authors

I propose still a few point for correction:

1.      The title is still long and hard to understand: Population Structure, Genetic Diversity and Selective Sweeps Candidate Genes for the adaptation in Picea koraiensis revealed by single nucleotide polymorphism (SNP) markers.

2.      Fig. 3 b needs correction like Fig. 3 a.

3.      In Fig. 5 all values are significant I still propose to rank from highest to lowest values for genes, because the figure is not so clear to read.

Yours sincerely,

Reviewer 3 Report

I recommend it to be published with minor revisions on the following aspects:

You answered in responce8 that you used the sliding window method, but you did not indicate in the article that you used the sliding window method.

If Fst and π are obtained using the sliding window method, I think you should include the criteria (window size and step size) in Materials and Methods.

I understand that you selected the candidate SNPs for selective sweep based on the Fst and π ratio results. 

However, I think you should provide criteria for the genes in the selected region used in the GO enrichiment analysis, as you don't seem to provide any criteria (how many bases within from the SNP or genes contained within the sliding window).
